# Detection of Collaterals from Cone-Beam CT Images in Stroke

**DOI:** 10.3390/s21238099

**Published:** 2021-12-03

**Authors:** Azrina Abd Aziz, Lila Iznita Izhar, Vijanth Sagayan Asirvadam, Tong Boon Tang, Azimah Ajam, Zaid Omar, Sobri Muda

**Affiliations:** 1Department of Electrical and Electronic Engineering, Universiti Teknologi PETRONAS (UTP), Seri Iskandar 32610, Malaysia; lila.izhar@utp.edu.my (L.I.I.); vijanth_sagayan@utp.edu.my (V.S.A.); azimahajam@gmail.com (A.A.); 2Centre for Intelligent Signal and Imaging Research (CISIR), Department of Electrical and Electronic Engineering, Universiti Teknologi PETRONAS (UTP), Seri Iskandar 32610, Malaysia; tongboon.tang@utp.edu.my; 3School of Electrical Engineering, Faculty of Engineering, Universiti Teknologi Malaysia, Johor Bahru 81310, Malaysia; zaid@fke.utm.my; 4Department of Radiology, Faculty of Medicine and Health Sciences, Hospital Pengajar Universiti Putra Malaysia, Serdang 43400, Malaysia; asobri@upm.edu.my

**Keywords:** collaterals, cone-beam computed tomography (CBCT), stroke, support vector machine (SVM), K-nearest neighbors (KNN)

## Abstract

Collateral vessels play an important role in the restoration of blood flow to the ischemic tissues of stroke patients, and the quality of collateral flow has major impact on reducing treatment delay and increasing the success rate of reperfusion. Due to high spatial resolution and rapid scan time, advance imaging using the cone-beam computed tomography (CBCT) is gaining more attention over the conventional angiography in acute stroke diagnosis. Detecting collateral vessels from CBCT images is a challenging task due to the presence of noises and artifacts, small-size and non-uniform structure of vessels. This paper presents a technique to objectively identify collateral vessels from non-collateral vessels. In our technique, several filters are used on the CBCT images of stroke patients to remove noises and artifacts, then multiscale top-hat transformation method is implemented on the pre-processed images to further enhance the vessels. Next, we applied three types of feature extraction methods which are gray level co-occurrence matrix (GLCM), moment invariant, and shape to explore which feature is best to classify the collateral vessels. These features are then used by the support vector machine (SVM), random forest, decision tree, and K-nearest neighbors (KNN) classifiers to classify vessels. Finally, the performance of these classifiers is evaluated in terms of accuracy, sensitivity, precision, recall, F-Measure, and area under the receiver operating characteristics curve. Our results show that all classifiers achieve promising classification accuracy above 90% and able to detect the collateral and non-collateral vessels from images.

## 1. Introduction

Stroke is the third leading cause of mortality in Malaysia, and the cases of stroke are growing every year [1]. Stroke can damage brain tissues, and impair our body functions leading to numbness, weakness, or paralysis on the side of the body opposite to the obstruction of the artery. Collateral vessels are vessels that connect the damaged tissue with other arteries and restore the blood flow to the tissues, thus preventing serious damage and offer better chance of recovery. Imaging of the brain vessels has shown that collateral flow can sustain brain tissue for hours after the occlusion of major arteries to the brain, and maintenance of collateral flow is therefore important. Several studies show that the outcome after stroke treatment is highly dependent on the extent of collateral flow [2,3]. The number of collateral vessels varies from person to person, contributing to different stroke outcomes. A patient with good collateral flow may have favorable outcomes and better clinical response to a recanalization therapy, even though the treatment is initiated at delayed hours after the onset of stroke [4,5], unlike a patient with poor collateral flow.

Imaging plays a critical role in the diagnosis of stroke patients before initiating treatment and for early intervention. There are various imaging techniques available for assessing vascular lesions and brain tissue status in acute stroke patients. Digital subtraction angiography (DSA) is gold standard imaging to identify collateral vessels in acute stroke. However, DSA only provides two-dimensional (2D) imaging, and it is an invasive technique requiring a special contrast medium or ‘dye’ to be injected into blood vessels to make the vessels visible. Imaging techniques based on computed tomography (CT) and magnetic resonance imaging (MRI) are able to provide non-invasive angiography and perfusion imaging for the identification of cerebral injury in the first few hours after arterial occlusion [6]. But these imaging techniques have low spatial information of collateral vessels.

Cone-beam computed tomography (CBCT), on the other hand, is an advanced imaging modality that provides accurate and three-dimensional (3D) imaging of hard tissues, soft tissues, and bones. Recently, CBCT is increasingly popular in acute stroke and neurovascular image-guided procedures due to its advantages over conventional CT, such as less x-ray beam limitation, high spatial resolution and rapid scan time depending on the manufacturer providers [7,8]. A newer study applying machine learning techniques to automate the process of estimating stroke tissues has shown a promising result in accuracy [9]. However, this work uses MRI images as supposed to CBCT images. To the best of our ability, we could not find any work on the detection of collateral vessels for stroke cases using CBCT imaging modality based on image processing approach. The only closely related work is about the vessel detection of CBCT images on prostatic arteries [10].

Despite many advantages of CBCT, there are a number of challenges in detecting collateral vessels from CBCT images which are: (1) low contrast, (2) existence of various noises and artifacts such as motion, beam hardening, aliasing and ring artifacts, and (3) non-uniform illumination caused by exponential edge gradient effect [11]. In addition, it is challenging to detect collateral vessels in brain due to their randomized and non-homogeneous structure, and small size (ranging from 0.4–0.6 mm [12]).

Image processing is a popular technique used for detection and classification of various diseases. The work in [13] uses the hyperspectral imaging technique to detect blood vessels and to differentiate the artery from vein during surgeries using MRI images. Several works are proposed for detecting blood vessels of retinal images based on fundus modality. The complete review on these works is available in [14]. In our work, we use image processing technique for the application of stroke and our images are based on CBCT image modality.

In our work, we develop a technique for accurate detection of collateral vessels from CBCT images to assist neuroradiologists to objectively visualize the collateral vessels. Typically, subjective assessment is used in diagnosis and depending on neuroradiologist’s expertise and experience, it may lead to different interpretation, consequently influencing the outcome of treatment received by patients. The proposed technique consists of: (1) image pre-processing, (2) feature extraction, and (3) classification processes. Firstly, the original images are processed and enhanced to improve their visibility. Then, we extract their texture and shape features. Finally, we investigate several types of classifiers on the extracted features to explore which classifier is the best to classify the collateral vessels from normal vessels. The performance of our detection technique is then validated using a 25% holdout cross-validation technique and measured using various performance metrics such as accuracy, sensitivity, precision, recall, F-Measure, and AUC values of the receiver operating characteristics (ROC) curves.

The paper outline is as follows. In Section 2, we provide the review on the related works for detecting the collateral vessels. Next, Section 3 presents the methodology of our work, and Section 4 discusses results and presents the analysis of the results. Lastly, Section 5 concludes the paper and suggests further work in this area.

## 2. Related Works

Blood vessel detection based on image processing typically consists of various processes ranging from artifact removal, vessel enhancement to feature extraction. We discuss related works on these processes in the following section.

### 2.1. Artifact Removal

Artifact removal is a vital pre-processing step to remove unwanted artifacts from affecting the interpretation of the analyzed image. In recent years, many researchers have been developing artifact removal methods for the CBCT images, especially ring artifact in human abdomen, brain and dental applications [7,15,16,17,18,19,20].

Artifact removal can be divided into two categories: (1) pre-processing in the projection domain, and (2) post-processing in the image domain. The methods underlying the projection domain include the calibration of the cone-beam source and detector during the acquisition process [18]. These methods modify the projection angle and the cone-beam geometry to remove metal artifacts in the CBCT image. Since we cannot modify the parameters during image acquisition and raw projection of CBCT, we opt to apply our study in the CBCT DICOM image. Ibraheem et al. [7] used a dental image to reduce artifact to improve three-dimensional reconstruction. Their artifact reduction includes a contrast adjustment at initial stage to improve the contrast of bone, soft tissue, and background. They used a double threshold and closing algorithm to the contrast-adjusted image to determine the noisy area and other organs. They perform Otsu’s threshold and boundary tracing, where it labels the pixels’ value with zero pixels representing the background. The algorithm was applied to the reconstructed image using radon transformation and finally displayed the image in a sinusoidal graph from 0 to 360 degrees. The noisy data was removed during thresholding on the radon transform image. The image is then back-projected to display the processed image.

Yilmaz et al. [21] compared several methods such as anisotropic diffusion filter, block-matching and 3D filtering (BM3D), and Wiener filter to reduce noise and artifact in CBCT image. They found that the conventional anisotropic diffusion filter in [22] may reduce noise in the CBCT image by smoothing the image while preserving the edges. Later, Yilmaz et al. [17] introduced an adaptive anisotropic diffusion filter to improve the conventional method by adding a blur metric as an iteration stopping criterion [23]. The performance of the adaptive approach towards the CBCT image was measured using peak-signal-to-noise ratio (PSNR), mean absolute error (MAE), and blur level (blevel) score. The adaptive method proposed in [17] showed better performance compared to the performance of conventional AD filter for three types of noise, which are Gaussian, Speckle, and Poisson noise. The conventional anisotropic diffusion filter [22] may smooth the image and remove noise, but it will distort the information in the image, especially small structures like a collateral vessel.

The authors in [16,24] develop an algorithm to remove artifact in CBCT image by using smoothing techniques called L${}_{1}$ and L${}_{2}$ smoothing and this technique is inspired from [25]. This is a popular method for ring artifact removal, where it involves transformation from cartesian to polar coordinate systems [26,27]. Here, the ring artifacts are manifested as stripe artifacts in polar coordinates and after processing the image in polar coordinates, it is transformed back to the cartesian coordinates. However, this smoothing method is not suitable for collateral vessel images which are tiny in nature. The application of smoothing on collateral vessels may eliminate the low-amplitude structures.

Considering the small-size collateral vessels, we investigate various filtering techniques for the vessels such as Median filter, Wiener filter, L${}_{1}$ and L${}_{2}$ relative total variation (L${}_{1}$-RTV and L${}_{2}$-RTV, respectively) with the objective to find the best technique capable of extracting the collateral vessels. Median and Wiener filters are conventional methods that are widely used in image filtering, meanwhile L${}_{1}$-RTV and L${}_{2}$-RTV are used for both structure and texture extraction. In L${}_{1}$-RTV and L${}_{2}$-RTV, smoothing functions L${}_{1}$ and L${}_{2}$ are first used to extract the wanted texture from the smoothed image by subtracting the smoothed image from the original image, then RTV is applied on the texture to extract the structure.

### 2.2. Vessel Enhancement

The basic idea of vessel enhancement is to increase the contrast between the bright (vessel) and dark (background) regions of the image. The Hessian-based filter is the most popular method of vessel enhancement which has been used for a decade in many medical imaging applications [28,29,30]. The Hessian-based filter is suitable for multiscale analysis, especially in blood vessels enhancement of the diameter that forms in a multiscale range. The Hessian-based Frangi filter is useful in vessel enhancement, but it is susceptible to noise and tends to create a vessel-like object from the noise [31]. This effect happens during the artifact and noise removal process in which the artifact and noise suppressed in the background generated a non-uniform illumination imposing a challenge in vessel extraction.

A non-linear filtering method based on mathematical morphology [32] that combines two basic morphological operations (erosion and dilation) can be employed to address the drawback of Hessian-based Frangi filter. One of the morphological techniques is known as top-hat transformation and it can deal with a non-uniform background illumination [33]. However, the classical top-hat is suitable for high signal-to-noise ratio (SNR) images, and in the case of low SNR, its performance is poor [34]. Sun et al. [33] use a top-hat enhancement to normalize the background as vessel images may have a non-uniform background. By normalizing the background image, the filtering of the vessel structure is improved, as well as the segmentation process. The crucial part of the top-hat transformation is to find the appropriate structuring element that may improve the performance of top-hat transformation [34]. Therefore, Bai et al. [34] introduce a new technique for finding the appropriate structuring element with the aim of detecting a small target in the infrared dim image. They used two different but correlated structuring elements to differentiate the target from surrounding regions.

Most of the Hessian-based approach relies on image intensity. It may lead to poor enhancement, or it may suppress the finer and lower intensity vessels. Phase Congruency Tensor-based approaches on the other hand are image contrast-independent [32]. Local tensor can detect structures in any direction. By taking advantage of both top-hat transformation and local tensor methods, Alharbi et al. [32] combine these two methods and propose a technique called Multiscale Top-Hat Tensor (MTHT). MTHT shows better enhancement performance compared to the performance of Hessian-based and top-hat transformation alone. Due to its promising result, we will explore this method in our work to enhance the small vessels such as collateral vessels.

### 2.3. Feature Extraction

The feature extraction aims to characterize pixels in terms of some quantifiable measurements to identify whether the pixels are normal or collateral vessels.

*Texture-based Feature Extraction*: Many techniques have been used to extract features from images, and one of them is the texture feature [35]. In general, texture refers to surface characteristics and appearance of an object given by the size, shape, density, arrangement and the proportion of its elementary parts [36]. Due to the significance of texture information, texture-based feature extraction is a crucial function in various image processing applications like remote sensing, medical imaging, and content-based image retrieval [37]. The primary purpose of texture-based feature extraction is to map differences in spatial structures, either stochastic or geometric, into differences in gray value [38]. First-order statistical feature based on histogram of gray level intensity and the second-order statistical feature based on gray level co-occurrence matrix (GLCM) are two important methods of texture-based feature [39,40].

Another texture-based feature extraction method is a local binary pattern (LBP). The LBP features are also important to express the image texture analysis where it encodes local texture information around each pixel [37,41]. Each central pixel is compared with 4, 8, 12, 16, or 24 neighborhood pixels. If the neighboring pixels are smaller than the central pixel, then the central pixel will become zero [42]. LBP features have been used in face recognition [43] and histopathology images [44]. Gray-level co-occurrence matrix (GLCM) is a second-order statistical texture analysis method can reveal certain properties about the spatial distribution of the gray levels in the texture image [39,40,45].

*Shape-based Feature Extraction*: Other than texture features, shape features can be used to obtain meaningful information from an image. Yang et al. [46] survey the shape-based feature extraction method, where the author divides the shape features into six categories. In one of those categories, the shape features can be divided into two approaches, which are contour-based and region-based descriptors [47]. The basic descriptor for the region-based method employed geometric features. The geometric features use the information from a morphological operation such as area, compactness, minor, and major axis length to derive five shape factors [48].

*Contour-based Feature Extraction*: The contour-based technique can be represented by the invariant moments. It is a type of statistical descriptor for shape features that were first introduced by Hu [49]. The invariant moments describe a shape’s layout, which is the arrangement of its pixels or a bit like combining area, compactness, irregularity, and higher-order descriptions together [50]. The basic formula from [51,52] was widely used to calculate the invariant moments.

In this paper, we use GLCM, shape and moment invariant to evaluate the performance of these features in segregating the images into their significant classes.

### 2.4. Classification

Classification classifies images into several categories based on their similarities. Examples of popular classification methods are k-nearest neighbor (KNN), support vector machine (SVM), and decision tree, and they use statistical classifier. The KNN consists of finding a fixed number of elements from the training set, which is the nearest to the test sample according to a given distance function [53]. The SVM aims at learning the parameters of a hyperplane in the feature space, optimally separating the elements of the training set in the different classes [54]. A decision tree is a tree-based classifier built from the training set, where leaf nodes correspond to the classes, and internal nodes are feature condition points that divide instances with different characteristics [55]. Convolutional neural network (CNN) is used to detect pulmonary vascular diseases [56] and classify the artery fundus images to assess diabetes, hypertension or other cardiovascular pathologies [57].

Amelio et al. [55] survey the classification methods for image-based medical analytics. KNN and SVM are the mostly used supervised classifiers for medical imaging to identify the type of disease, veins or blood vessels. In the previous studies [58,59], the authors use the SVM classifier for Alzheimer’s disease diagnosis. In [58], the classifier is trained by the image represented by the local binary pattern (LBP). In a similar case, Xiao et al. [59] extract textural and region-based local features as the input for the SVM classifier. The SVM and KNN classifiers are also used in the esophageal X-ray images to identify different types of cancerous images [60]. In our work, SVM, decision tree, random forest, and KNN classifiers are used to classify vessels into collateral and non-collateral (normal) vessels.

## 3. Methodology

The chart in Figure 1 illustrates the collateral vessel detection methodology in our study.

### 3.1. Image Acquisition

In this study, we use anonymous data of the CBCT image validated by radiologists. The images are acquired from Hospital Universiti Kebangsaan Malaysia (HUKM) using a Philips X-ray scanner. CBCT image is a 3D volumetric image acquired using a flat detector of the angiography system, which reconstructed as axial CT like images. The images obtained of stroke patients used are in the DICOM format. The thickness of each image slice is 0.762841 mm, and the image size is 256 × 256.

### 3.2. Conversion of CBCT to CT Values

CT image is reconstructed based on the attenuation coefficient, whereby air, water, soft tissues, and bone have their attenuation value. The attenuation coefficient measures how easily the tissue can be penetrated by an X-ray beam. The gray value in CT image is called CT value and it is measured in the Hounsfield unit (HU). CBCT image has a larger amount of scattered X-rays that resulted in artifacts. Scatter is well known to further reduce soft-tissue contrast and it will also affect the density values of all other tissues [Artifact: the downturn of CBCT image]. and the Different from CT image, CBCT, as mention in the previous section, has a larger amount of scattered X-rays. Because of the scatter and resultant artifact, CT values in CBCT are not valid [61]. In the previous study [62], a linear relationship between gray values in CBCT and the attenuation coefficient of various materials in several CBCT scanners was reported [63]. It can be concluded that the conversion of gray values from CBCT to CT values could be performed. In another study, Chindasombatjaroen et al. [63] highlighted that the pixels value in CBCT is lower than CT values in CT. Therefore, we proposed two steps in the CT value conversion of CBCT image; (1) brightness adjustment and (2) CT value conversion.

*Brightness adjustment*: The aim is to adjust the tonal range of the image to increase the intensity of blood vessels in the image. We adjust the brightness of the input image by implementing a linear regression model, as described in Equation (Equation 1). The regression model in Equation (Equation 1) is designed using a quality factor, *Q*, as a predictor variable, and recognition performance, Y, as the response variable in the model [64],
(1)Y=m+wQ
where *w* represents the corresponding regression coefficients, and *m* is the regression constant.

*CT value conversion*: During image acquisition, various rescale slope and rescale intercept are adopted in CT modality due to various manufacturing providers. It will affect the CT values in image reconstruction of CT or CBCT images. Therefore, CT value conversion aims to calculate the adjusted image, Y, in step 1 to obtain the CT value, HU, ensuring this converted image is capable of preserving as much information as possible, particularly the small vessel structures after the CT value conversion using Equation (Equation 2) [65]. The CT value conversion is performed as follows:(2)$$\mathrm{HU}=Y*R_{S}+R_{I}$$
where $R_{S}$ and $R_{I}$ are generally the rescale slope and rescale intercept, respectively. These values can be found in the DICOM file itself (i.e., image tag), numbered as and respectively. If the image tags are not present in the DICOM file, it means that the CT values are in the default unit, which is Hounsfield (HU) unit. The brightness adjustment and conversion of CT value must then be performed for each image slice.

### 3.3. Maximum Intensity Projection

Maximum intensity projection (MIP) is subsequently performed onto the converted images, HU, to extract pixels with the highest Hounsfield value along the Z-axis, as shown in the diagram in Figure 2a, so that in a single bi-dimensional image all dense structures in a given volume are observed. The aim is to create an image similar to the ground truth provided by the radiologist using a HOROS software since the MIP method tends to display bone and contrast material-filled structures preferentially. Therefore, the MIP is the most suitable approach compared with minimum and average intensity projection. Figure 2a is the example of 16 mm thick-stacked images and Figure 2b is the result of the MIP process. The size of thickness is selected to replicate the ground truth images prepared by the expert. Using stacked images in MIP, we can improve the visibility of the small vessels in the image compared to only relying on one single slice image. This process will allow us to extract as much information as possible from the image such as the visualization of a long segment of a vessel, decreased perceived image noise as well as providing a sufficiently high contrast between vessel of interest and surrounding structures. In medical practice, the clinicians use MIP images to differentiate the normal blood vessels from the collateral blood vessels.

The MIP step is performed in every slice. Assuming one subject has 256 sliced images, the first MIP image is from slice 1 to 20. With a gap of one slice, the second MIP image will be from slice 2 to 21. This process will go on until it reaches the last image slice. This is illustrated in Figure 3.

### 3.4. Skull Removal

The purpose of skull removal is to obtain an MIP image without the skull to facilitate collateral vessel enhancement and segmentation. A mask image of the region of interest (brain area without the skull) is obtained for the skull removal to avoid any false-positive results since the intensity range of the skull is quite close to the intensity range of blood vessels. Skull removal algorithm based on morphological reconstruction, which relies on the connected components and region properties, is used. Skull will be removed based on the labeling connected components and computation of region properties. Figure 4 shows the resultant images obtained throughout the skull removal process to obtain blood vessel structures from an MIP image.

The skull removal involves several processes. MIP image obtained from Section 3.3 is used as the input image for this purpose. The first process is to convert the input image into a binary image. To achieve this, we use a thresholding method (I<x) where *x* is a scalar value that is obtained from a mean value of input image, *I*. The result is as illustrated in Figure 4b. The second process is to eliminate unwanted region; in this case, the white region surrounding the outer area of the skull in Figure 4b. This area is also known as the area connected to the image border. Hence, to remove this, we use the method that removes/clear out all connected components or pixels connected to the image boundary which are the outer white pixels surrounding the skull region. The result image obtained is illustrated in Figure 4c.

The third process is to create a mask image with only white pixels in the foreground and black pixels in the background. The process involves a flood-fill operation that will change connected background pixels (0 s) to foreground pixels (1 s) and a measurement of area for each region. Then, by selecting the largest area, what left is the brain region as illustrated in Figure 4d; the mask image. Lastly, the final process is to obtain the MIP input image without the skull. By performing an element-by-element multiplication involving the MIP input image and the mask image, we obtain the final result as illustrated in Figure 4e.

### 3.5. Noise and Artifact Removal

There are many filtering methods available for noise and artifact removal in image processing. Before further processing and analysis, removing the noise and artifact is vital to obtain an accurate result. From previous studies in [17,20], the anisotropic diffusion filter has the ability to smoothen images while preserving the edges, thus we use this filter to remove noise in the CBCT images. In our case, the anisotropic diffusion filter may cause the loss of valuable information on the blood vessels’ structure during the smoothing process, as shown in Figure 8. The peak signal-to-noise ratio and structural similarity index are used to measure the image quality, calculated using Equations (Equation 3)–(Equation 5).
(3)$$\mathrm{PSNR}=10*log(\frac{R^{2}}{MSE})$$
(4)$$\mathrm{MSE}=\frac{\sum {\left[I_{1}\left(m,n\right)-I_{2}\left(m,n\right)\right]}^{2}}{M*N}$$
(5)$$\mathrm{SSIM}=\frac{\left(2\mu_{x}\mu_{y}+c_{1}\right)\left(2\sigma_{xy}+c_{2}\right)}{\left(\mu_{x}^{2}+\mu_{y}^{2}+c_{1}\right)\left(\sigma_{x}^{2}+\sigma_{y}^{2}+c_{2}\right)}$$
where $c_{1}={\left(K_{1}L\right)}^{2}$ and $c_{2}={\left(K_{2}L\right)}^{2}$. *L* is the dynamic range of the pixel value. The constant $c_{1}$ and $c_{2}$ are used to avoid instability for image regions where the local mean or standard deviation is close to zero. Therefore, a small non-zero value of constant $c_{1}$ and $c_{2}$ are calculated by setting $K_{1}=0.01$, and $K_{2}=0.03$. The parameter values can be referred to the work in [16]. The results of PSNR and SSIM values are depicted in Table 2.

### 3.6. Vessel Enhancement

The final preprocessing step consists of generating a new vessel enhanced image for feature extraction. Vessel enhancement is performed by applying the morphological top-hat transformation. It involves several morphological operations, which are dilation, erosion, and opening. Let f(x,y) and B(u,v) represent the grayscale image and the structuring element, respectively. The dilation and erosion of f(x,y) and B(u,v) are denoted by f⊕B and f⊖B respectively and they are defined by the following equations: (6)$$f⊕B(x,y)=\max_{u,v}\left(f\left(x-u,y-v\right)+B\left(u,v\right)\right)$$
(7)$$f⊖B(x,y)=\max_{u,v}\left(f\left(x-u,y-v\right)+B\left(u,v\right)\right)$$

Based on dilation and erosion operations, opening and closing of f(x,y) and B(u,v), indicated as f∘B and f•B are defined as follows: (8)f∘B(x,y)=(f⊖B)⊕B
(9)f•B(x,y)=(f⊕B)⊖B

The opening operation is used to smoothen bright small regions of an image, while the closing operation eliminates small dark holes. There are white and black top-hat (called bottom hat) which are denoted by $I_{WTH}$ and $I_{BTH}$,
(10)$$I_{WTH}=f\left(x,y\right)-\left(f\circ B\right)\left(x,y\right)$$
(11)$$I_{BTH}=\left(f•B\right)\left(x,y\right)-f\left(x,y\right)$$
where γ is a morphological opening operation using a disc structuring element of ten pixels in radius. Thus, while bright vessel structures are removed, the darker structures remaining after the opening operation become enhanced, for example, blood vessels. This method can extract vessels in non-uniform illumination. The sample result of vessel enhancement operation is shown in Figure 8a. Then, a contrast adjustment is applied to increase the contrast of the blood vessel shown in Figure 8b.

The images used in this study have a bright foreground and dark background, hence the suitable method used is white top-hat transformation. The result of the top-hat transformation illustrated in Figure 8a.

### 3.7. Feature Extraction

In this study, we have selected three types of feature extractors for characterizing the non-collateral and collateral case as summarized in Table 1. The extracted features are evaluated and analysed to identify the features that can segregate normal and collateral vessels into their classes in the best manner. In total, 40 features per image were extracted.

*Sliding window*: Sliding window method is another name for block processing. There are two types of block processing approaches; non-overlapping and overlapping. The concept of non-overlapping is by means of dividing an image into a uniform size of blocks. The disadvantage of a non-overlapping block is that the object that we want to extract might be and might not be inside the block or the object separated into different blocks. It will lead to a misinterpretation of the object; for example, normal vessels are interpreted as collateral and otherwise.

The overlapping block processing method on the other hand is more suitable in this case to avoid misinterpretation. In this method, as for example, if the block size is 16 with a pixel shift by 1, then the first block will be the indices of 1:16 in *x* and *y* while the second block will be 2:17, and the third block will be 3:18, and so forth. This means that there is an overlap data between neighboring blocks. So, in this case, the number of blocks will be multiplied. However, the drawback of the overlapping block is it is time-consuming depending on the number of pixel shift selected. Figure 5 depicts an image of a sliding window for overlapping blocks. The red square is the window that will move until the very end of the pixel available in the image. In our work, all feature information ranging from texture and shape-based are extracted from each image patch obtained using the overlapping block sliding window method.

### 3.8. Classification

Four classification algorithms which are support vector machine (SVM), random forest, decision tree (DET) and K-nearest neighbor (KNN) are used to classify collateral and normal vessels based on the forthy features extracted in the previous process. The SVM in our work uses polynomial kernel of 3 degree hyperparameter. For the random forest, half of input features for a logistic model training is used to sample at each split point. For the decision tree classifier, a standard model is used with maximum of 100 splits. As for the KNN, we use a standardized model consisting of Euclidean distance function with equal-distance weights. The number of neighbors is set to 2. We select the K value by running the KNN algorithm many times with different values of K, The K value that is best to reduce the number errors while maintaining the accuracy is chosen.

The 10-fold cross validation is used to measure the performance of each classifier and to evaluate how well the classifier generalizes the data. A total of 100 data was used for the 10-fold cross validation meanwhile additional 25 data were used for testing after the model is trained.

### 3.9. Performance Measures

The performance measures used to evaluate and analyze the features and classifiers such as sensitivity, specificity, accuracy, precision, F-Measure, AUC and ROC-AUC curve are presented in this section.

The statistical calculations for the first three performance measures are given as follows,
(12)$$\mathrm{Sensitivity}=\frac{TP}{TP+FN}$$
(13)$$\mathrm{Specificity}=\frac{TN}{TN+FP}$$
(14)$$\mathrm{Accuracy}=\frac{TP+TN}{TP+TN+FP+FN}$$
where TP (True Positive) represents the number of pixels that are classified correctly as collateral vessels, TN (True Negative) represents the number of pixels classified correctly as non-collateral vessels, FP (False Positive) represents the number of pixels belonging to non-collateral vessels classified as collateral vessels, and FN (False Negative) represents the number of pixels belonging to collateral vessels classified as non-collateral vessels.

Another performance measure that is used which is the F-Measure, refers to the harmonic means of the precision and recall, which is calculated as below,
(15)$$F\text{-}\mathrm{Measure}=2*\frac{precision*recall}{precision+recall}$$
where recall refers to the percentage of total relevant results that are correctly classified, and precision refers to the percentage of the relevant results. Both are calculated as follows.
(16)$$\mathrm{Recall}=\frac{TP}{TP+FN}$$
(17)$$\mathrm{Precision}=\frac{TP}{TP+FP}$$

Besides that, we also measured the performance by using the area under the curve—receiver operating characteristics (AUC-ROC). It is one of the most important evaluation metrics for checking any classification model’s performance. It is used in this study to evaluate the quality of classifiers in correctly classifying the vessels into vessel and non-collateral vessel classes. The AUC-ROC curve plots two parameters; TPR (true positive rate) against FPR (false positive rate), to show the performance of the classifiers at all classification thresholds. The TPR and FPR can be defined as follows,
(18)$$\mathrm{TPR}=\frac{TP}{TP+FN}=Sensitivity$$
(19)$$\mathrm{FPR}=\frac{TP}{TP+FP}=1-Specificity$$

## 4. Results and Discussion

The results presented in this section are obtained using a dataset consisting of 125 MIP images obtained from CBCT images. Here, we also include results on feature evaluation for classification of vessels.

### 4.1. HU Conversion and MIP Projection

The regression model is used to estimate the regression coefficient to obtain the most promising image for further analysis. By implementing the linear equation in Equation (Equation 1), we can control the parameters of w and m to obtain the output that we desire. The linear function can be used to convert pixel values from the CBCT scanner to CT values using the Equation (Equation 2). As illustrated in Figure 6, if a CBCT image is converted to HU without adjusting the brightness component by multiplication with the output of Equation (Equation 20), we will lose the information of the blood vessels in MIP image.
(20)Y=1.0236Q+200
where *Q* indicates pixel value in CBCT.

MIP image (a) without brightness adjustment prior to HU conversion in comparison to (b) with brightness adjustment prior to HU conversion. As shown in Figure 6b, some vessel information is preserved as compared to that of Figure 6a. Therefore, it is necessary to perform brightness adjustment before converting the image into HU. The conversion is applied for each slice before performing the maximum intensity projection. Then, skull removal will be carried out to obtain MIP images without the skull as shown in Figure 4e.

### 4.2. Artifact Removal

Five noise filtering methods as given in Table 2, have been applied to the skull removed MIP images to reduce unwanted features/artifacts while smoothing the image. Figure 7 shows the resultant images after performing the noise filtering methods. The performances of the methods are compared based on quality metrics such as peak signal-to-noise ratio (PSNR) and structural similarity index measure (SSIM) as given in Table 2. In the ideal case, the highest PSNR value indicates that the quality of the image is the best as the distortion is the smallest. Similarly for SSIM, when its value is higher or converges to 1, the smaller is the image distortion of the reconstructed image in terms of brightness, contrast and structure respectively. As shown in Figure 7c, the image after Wiener filtering has the least distortion of vessel structure and contains less noise and artifact. However, as tabulated in Table 2, the Wiener filtering method has the lowest PSNR unlike its SSIM value which is the highest. It can also be observed that some details were also lost as a result of filtering using the L${}_{1}$ and L${}_{2}$ total variation smoothing methods as shown in Figure 7d and Figure 7e respectively. From these findings, we select the Wiener filtering method that achieved the best SSIM value for smoothed and noise reduced image.

### 4.3. Vessel Enhancement

Once image is smoothed and noise reduced by Wiener filtering method (see Figure 8a), vessel enhancement is performed to highlight further the vessel structures. To enhance further the vessels in non-uniform background, we employed top-hat transformation which is based on the residual; the difference between two or more common operators.

### 4.4. Feature Extraction

Once vessels are further enhanced, features are extracted from the vessels to be used as the input for classifiers to determine the type of vessels by the information given. The feature analysis in this study includes extraction of GLCM, shape, and invariant moments (refer Table 1). In feature extraction, first, we need to obtain image patches that contain either collateral or normal vessels from a filtered and enhanced MIP image e.g., Figure 8b, based on ground truth (labelled as collateral vessels). The aim is to create a library of features that stores all the feature information extracted from these patches.

Figure 9 is the zoomed-in version of the image of Figure 8b on the left hemisphere. The region of interest in our study is the blood vessels hence we marked several locations according to the ground truth image, as highlighted in red dots in Figure 9. These marked ROIs represent the location where image patches were obtained/cropped. The location of patches will be similar for all selected MIP images. In ground truth images, collateral vessels are delineated and labeled by the experts.

Samples of image patches obtained from three MIP images are depicted in Figure 10; the first two rows represent normal vessels, while the last two rows represent collateral vessels. It is observed that the brightest blood vessels represent normal vessels because it is denser compared to that of collateral vessels. However, some parts of collateral vessels have similar intensity as to that of the normal vessels. Furthermore, they may also have similar sizes. Hence, the challenge lies in the segregation of collateral and normal vessels that are similar in intensities and sizes. To do this, region descriptors or features need to be extracted from the vessels to provide description/information about the vessels according to its type. In this study, the features as given in Table 1 were extracted from the image patches obtained using sliding window approach (see Figure 5).

### 4.5. Performance of Classifiers

Table 3 and Figure 11 show the results of four classifiers for vessel classification performed on GLCM, Shape and moment invariant features with respect to their performance measures and AUC-ROC curve. Based on the average accuracy results, SVM is the best classifier to classify GLCM feature with 99.92% accuracy, meanwhile random forest is found to be best to classify the moment invariant with 78.04% accuracy, and finally KNN is more suitable the shape feature. We can also see that GLCM is the best feature for the CBCT image modality compared to momentinvariant and shape features in which the accuracy of GLCM is above 97% (97.41–99.92%) whereas the accuracy of moment invariant and shape is much lower which is about 77% with the highest is 79.33%. In terms of sensitivity and specificity, all four classifiers that use GLCM feature have values above 99% except for KNN classifier which obtains slightly lower value about 98.46% sensitivity and 96.40% specificity respectively. Similar trend is also observed for F-measure and AUC value for all classifiers applied on GLCM feature. They perform well when applied to GLCM compared to moment invariant and shape features. Since the AUC is related to the performance of sensitivity and specificity, the SVM for GLCM feature is expected to be the highest among all other classifiers which is 0.9999. SVM also produces the highest AUC value under moment invariant feature although its accuracy is not the highest one under this feature category. In shape feature, KNN has the best AUC value of 0.7964.

We can conclude our finding that for GLCM feature, SVM can provide better predictive ability in generating hyperplane that can optimally separate the collateral from normal vessel classes and GLCM is the best feature for our images despite the small data size.

Figure 11 shows the AUC-ROC curves of each type of features plotted for all four classifiers. We can observe that all features achieved good measure of separability as the AUCs are near to 1. It shows that the classifier models are capable of classifying the collateral and normal vessels into their classes based on these features.

In general, the performance of the classification models varies according to the type of features applied to the models. GLCM is found to be the best feature in our study. We can say that all the classifier models obtain satisfying results in using each feature to distinguish between collateral vessels and non-collateral vessels with SVN outperforms the rest of classifier for GLCM feature, followed by random forest for moment invariant and KNN for the shape feature. The results of our work is also consistent to the No Free Lunch (NFL) Theorem [66] which claims that the performance of classifiers varies according to the types of data set used.The computation time of our proposed work measured from pre-processing until classification is 35 seconds and it is computationally fast in comparison with manual detection technique performed by the clinicians.

## 5. Conclusions

Detection of collateral vessels in the management of acute ischemic stroke is crucial to assist clinicians in determining the type of treatment suitable for patient. This paper presented the MIP image construction from the CBCT images followed by pre-processing of the MIP images to distinguish vessel network from the background. In noise removal and image smoothing, Wiener filtering method has the highest SSIM value and is found to outperform other filtering methods such as anisotropic diffusion filter, median filter and RTV. However, the PSNR value obtained by this method is the lowest compared to the PSNR value of other filtering methods. Despite the lowest PSNR value, the Wiener filtering can preserve the small vessel structures with low distortion and reduce the noise that causes the non-uniform background. Further enhancement and background removal is achieved by means of multiscale top-hat transformation. Then, important features from several feature descriptors such as GLCM, shape, and moment invariant were extracted to analyze and represent the characteristics of collateral and non-collateral vessels. Altogether, about 40 features were extracted from 125 images. These features serve as the input for the classifiers to determine the type of vessels. In this study, four different classifier models were evaluated and compared namely SVM, KNN, decision tree and random forest. The classifiers were validated by the 10-fold cross-validation strategy running on 100 images and 25 images for testing. The performance for each feature type and classifier model was evaluated in terms of accuracy, sensitivity (recall), specificity, precision, F-measure and AUC values. Our findings show that the best feature for our CBCT image modality is GLCM whereas the SVM is the best classifier for this feature.

Since our proposed detection technique is based on conventional image processing, we anticipate that our technique may work on other medical image modalities such as CT and Magnetic Resonance Angiogram (MRA). Further investigation on the application of the proposed collateral detection on these modalities is our future interest. For future work, we will gather more data and explore deep learning techniques on our new data.

## Figures and Tables

**Figure 1 sensors-21-08099-f001:**
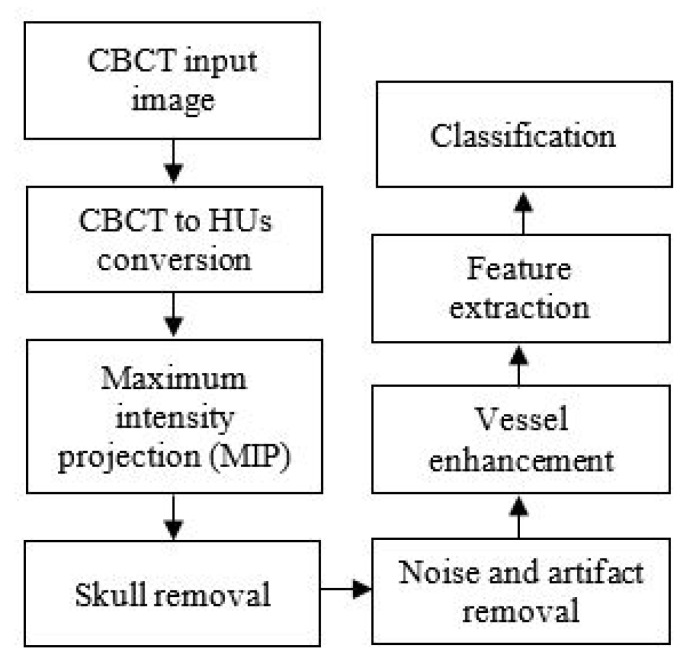
The flowchart of the proposed collateral vessel detection.

**Figure 2 sensors-21-08099-f002:**
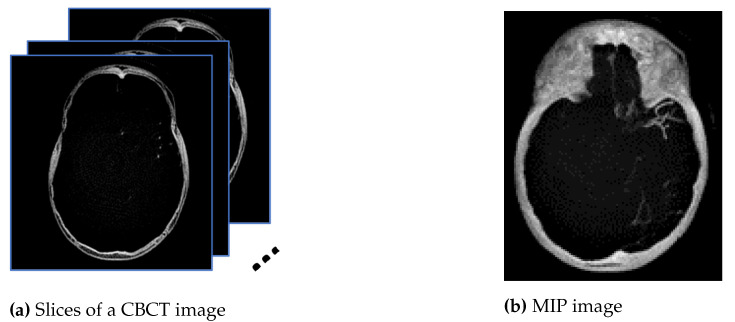
The images from (**a**) are stacked, and the maximum pixel value is extracted to generate the image in (**b**).

**Figure 3 sensors-21-08099-f003:**
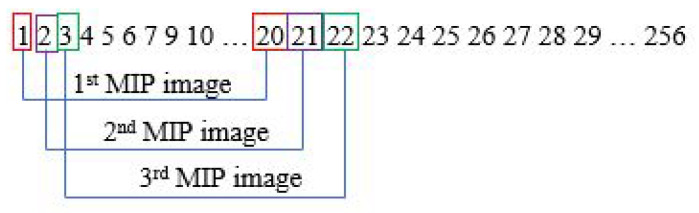
Steps for generating MIP images.

**Figure 4 sensors-21-08099-f004:**
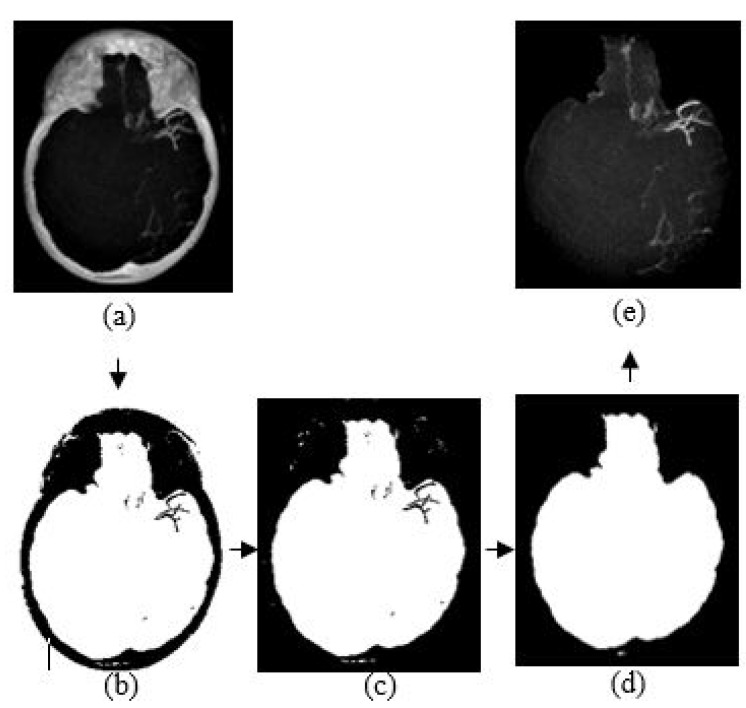
The algorithm of skull removal. (**a**) MIP input image, (**b**) binary image, (**c**) removing object touching the binary image boundary, (**d**) mask image, and (**e**) region of interest, masked out from mask image.

**Figure 5 sensors-21-08099-f005:**
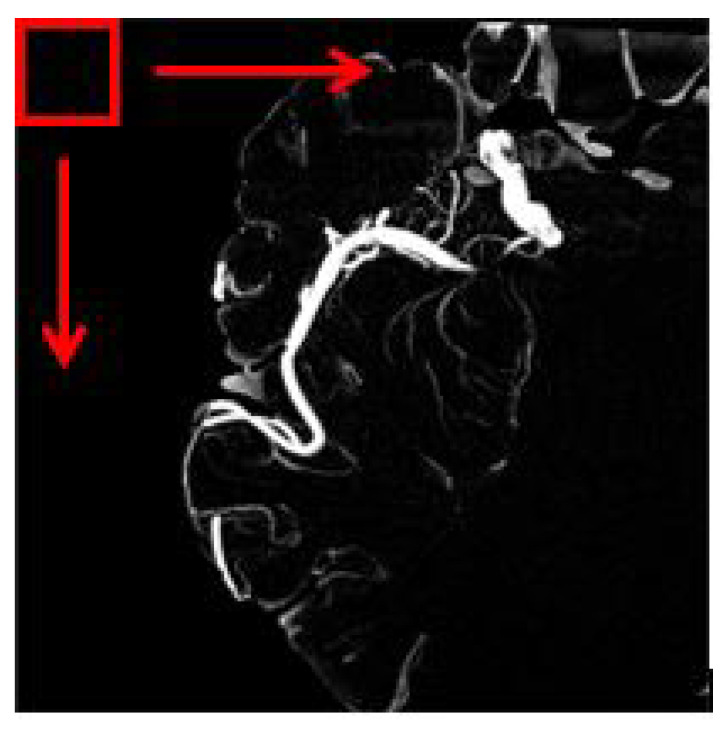
The red square represents the windowing slide.

**Figure 6 sensors-21-08099-f006:**
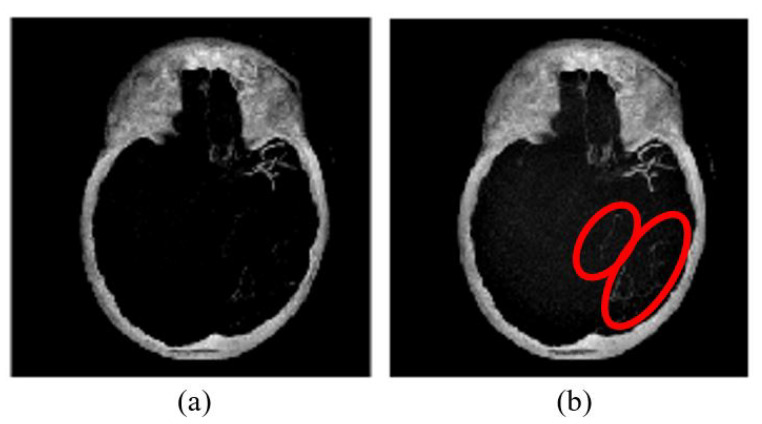
MIP image (**a**) without brightness adjustment prior to HU conversion in comparison to (**b**) with brightness adjustment prior to HU conversion.

**Figure 7 sensors-21-08099-f007:**
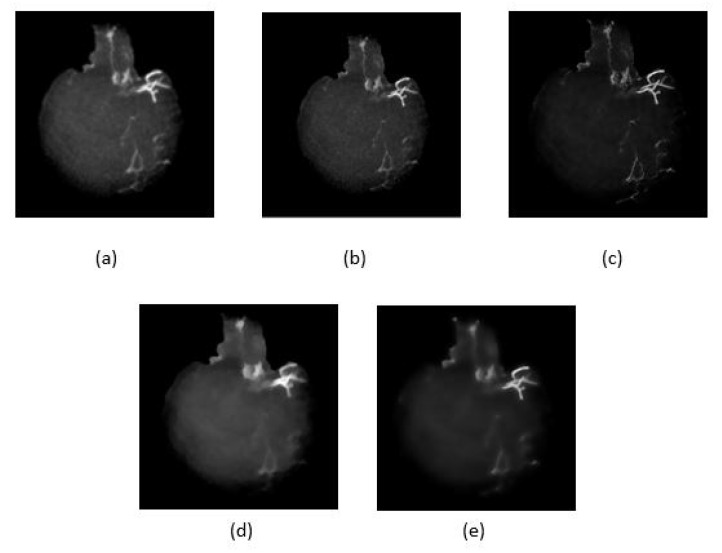
Five noise reduction methods used, (**a**) anisotropic diffusion filter, (**b**) Median filter, (**c**) Wiener filter, (**d**) and (**e**) are L${}_{1}$ and L${}_{2}$ total variation smoothing, respectively.

**Figure 8 sensors-21-08099-f008:**
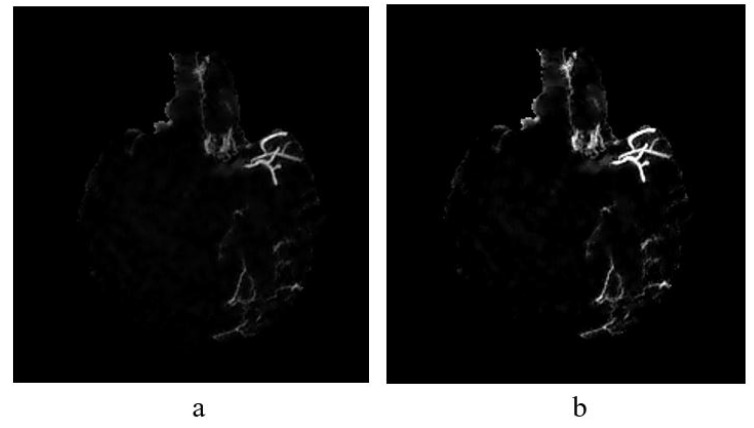
(**a**) The result of implementing top-hat transformation on the Wiener filtered image and (**b**) Contrast adjustment of top-hat transformed image.

**Figure 9 sensors-21-08099-f009:**
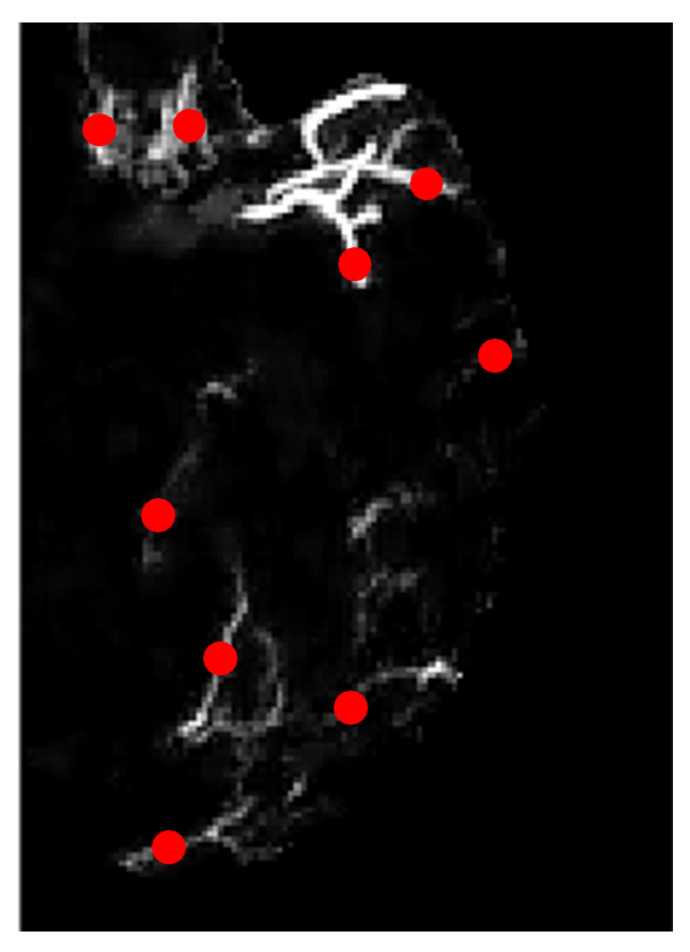
Zoomed-in image with marked locations in dotted red.

**Figure 10 sensors-21-08099-f010:**
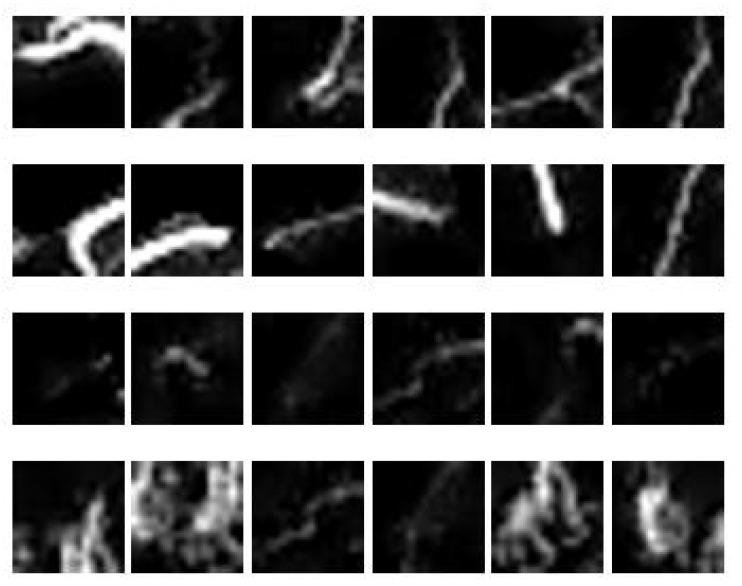
A sample of 24 image patches obtained from 3 MIP images.

**Figure 11 sensors-21-08099-f011:**
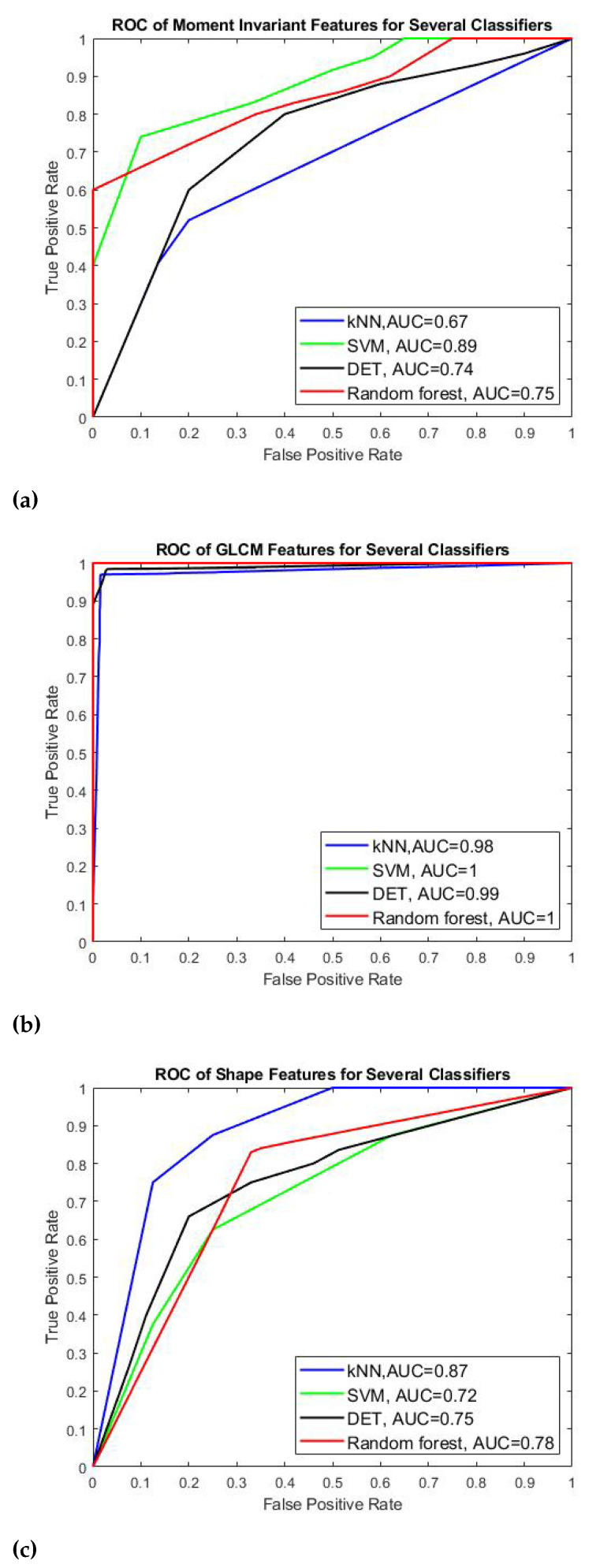
AUC-ROC curves for (**a**) Moment invariants, (**b**) GLCM, and (**c**) shape features.

**Table 1 sensors-21-08099-t001:** Feature Extraction Techniques.

Features	Name	Methods
Texture	GLCM	Haralick Texture Feature [39], Soh et al. [40], and Clausi et al. [45]
Shape	SHAPE	Shape factors using region properties [48]
	MOMENT	The seven Hu moment invariants [52]

**Table 2 sensors-21-08099-t002:** Quality metrics.

Method	PSNR	SSIM
Anisotropic diffusion filter	81.6645	0.9113
Median filter	82.7276	0.9112
Wiener filter	72.1510	0.9133
L${}_{1}$-RTV	78.0170	0.8603
L${}_{2}$-RTV	79.1176	0.8714

**Table 3 sensors-21-08099-t003:** Classification performance of four feature types.

Feature	Model	Accuracy (%)	Sensitivity (%)	Specificity (%)	Precision (%)	F-Measure (%)	AUC
GLCM	DET	99.89	99.87	99.92	99.92	99.89	0.9990
	SVM	**99.92**	99.87	**99.98**	**99.98**	**99.93**	**0.9999**
	Random Forest	99.85	**99.96**	99.75	99.75	99.85	0.9986
	KNN	97.41	98.46	96.40	96.31	97.37	0.9743
MOMENT INVARIANT	DET	77.67	79.01	76.70	75.67	77.19	0.7785
	SVM	77.58	69.21	**99.33**	**99.58**	81.64	**0.8427**
	Random Forest	**78.04**	69.83	98.07	98.75	**81.81**	0.8395
	KNN	68.33	75.65	64.37	54	62.89	0.7001
SHAPE	DET	71.5	72.16	71.12	70.16	71.03	0.7164
	SVM	72.20	71.07	74.02	75.91	73.23	0.7254
	Random Forest	78.41	76.18	81.63	**83.75**	79.64	0.7890
	KNN	**79.33**	**77.14**	**82.14**	83.66	**80.21**	**0.7964**

## Data Availability

Not Applicable.

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
