# Peer review of "Detection of Collaterals from Cone-Beam CT Images in Stroke"

_sensors, 2021, doi:10.3390/s21238099_

Round 1

Reviewer 1 Report

In this study, the authors propose an algorithm for detecting collateral vessels crucial to acute ischemic stroke. The work evaluated several methods for image reconstruction and pre-processing (filtering) to eliminate background noise and extract the essential features to distinguish collateral from non-collateral vessels. They also compared several machine learning classifier models with these features obtaining good performance metrics, including accuracy, sensitivity, specificity >80%. It is a promising study; there are only a few comments that need to be addressed.

  1. Could the authors comment on the generability of their detection model regarding the use of other x-ray scanners and angiography systems.
  2. Please indicate more clearly the percentage of images used to train and test the models and if an independent dataset from the latter was used for validating the classifier models using performance measures. From lines 457-458, it seems that the authors used a 5-fold cross-validation strategy and a dataset for validation of the classifiers that was not used for training, but this is not clear.
  3. In Methods: In lines 292-293, please provide a reference for "From previous studies, the anisotropic diffusion filter may remove noise in the CBCT image better than that of other methods" and explain why.
  4. Equations 13, 14, 15, 16, 17, 18, 19, and 20 in Section 4.5 Performance Measures calculations should be placed in the Methods Section.
  5. Could the authors comment on the time it takes from all the image analysis and algorithms calculations until the result is obtained: collateral vessel detection. This approach is intended to assist clinicians; therefore, it is essential to discuss this method's time-consuming.

Reviewer 2 Report

The paper presents a framework/methodology for the detection and segmentation of collateral vessels from CT/CBCT images. I have refered it as a framework as the authors have employed classic or basic image processing techniques for pre-procesing the images, but also in the high level stages, as segmentation.  Even so, it have been proven from the creation of the dataset till the evaluation of the results, that the performed work have a degree of success.

Nowadays, when CNN are employed on all tasks, I wonder why the authors haven't tried to improved their results by replacing kNN or DT with a CNN? Or even compare it. Also, the feature extraction methods are outdated. Even for the case of vessel processing, the classic morphology operators have been employed, or the connected components for skull removal, a more up-to-date feature extractor could be taken in account.

However, the paper is well written, good and logical structure, discussion and conclusion sections are  comprehensive, the performed research work is interesting and original. The only problem is the lack of scientific novelty, from the algorithmic point of view.

It is an honest work of the authors, I have no specific comments, as the decision for choosing some alternative methods have been proven. The paper could be of interest for readers as it is a return to classic image processing from the mighty CNN or other advanced Machine Learning.

Reviewer 3 Report

The present study presents an interesting topic and an image classification approach using feature extractors and machine learning algorithms. Despite the relevance of the theme, some aspects would significantly improve the quality of the paper.

The article structure is confusing. The “results and discussion” section presents information that should be in the “Methods” section. For example, the authors should describe the metrics in the methods section. I recommend separating the results and discussion as well. It will be much easier for the readers to understand the paper.

The introduction should provide more information to the readers regarding the state-of-the-art in image classification problems, medical image problems, and what has been done in this field.

Nowadays, deep learning is prevalent for image classification problems. For this reason, the authors should address this topic in the paper. The paper would present a substantial gain in quality and interest from the readers if including deep learning models in this evaluation. Similar works use deep learning methods. The description of the images in this paper (which already has ground truth) seems to be extremely suitable for applying these algorithms. Besides, deep learning methods are very good for feature extraction, and they might provide better results than the applied filters.

There is much room for a richer discussion. For example, how does this method advance in this field? How does it compare to other research in this field or broader areas such as medical imaging? How is this method compared to deep learning models in terms of performance? Is this method faster?

Why not experiment with more machine learning algorithms (random forest, XGBoost, etc.)?

The dataset should present a better description. For example, how many images were used? Besides, critical information is to clearly explain how the dataset was divided in training and testing. Besides, why did the authors not use a validation set to tune hyperparameters?

When using machine learning and deep learning models, it is essential to explain the hyperparameters used and how they were chosen.

Performance measure description should be placed in the methods section.

The sliding window method is present in many studies using deep learning. It would be interesting to include references on this theme and compare.

Figure 5 presents an error in the left side of the (b).

Figure 7 illustrates a semi-circle between images (a) and (b).

Figure 11 shows some strange results. For example, the authors should describe in depth why the ROC had 100% for some models. Besides, it is very strange for a ROC to present values below 0.5. The authors should investigate this.

Round 2

Reviewer 2 Report

The authors have responded generally to most of the issue raised by the first review round. As mentioned before, it is a honest work, with good results, using basic, classical image processing techniques, but gathered in a framework. The metrics employed suggest and sustains a good result.

Reviewer 3 Report

The authors significantly improved the paper quality.